# Neuroimaging Biomarkers in SCA2 Gene Carriers

**DOI:** 10.3390/ijms21031020

**Published:** 2020-02-04

**Authors:** Mario Mascalchi, Alessandra Vella

**Affiliations:** 1Department of Clinical and Experimental Biomedical Sciences, University of Florence, 50121 Florence, Italy; 2Nuclear Medicine at University Hospital, 53100 Siena, Italy; a.vella@ao-siena.toscana.it

**Keywords:** spinocerebellar ataxia type 2, magnetic resonance, brainstem, cerebellum, nuclear medicine

## Abstract

A variety of Magnetic Resonance (MR) and nuclear medicine (NM) techniques have been used in symptomatic and presymptomatic SCA2 gene carriers to explore, in vivo, the physiopathological biomarkers of the neurological dysfunctions characterizing the associated progressive disease that presents with a cerebellar syndrome, or less frequently, with a levodopa-responsive parkinsonian syndrome. Morphometry performed on T1-weighted images and diffusion MR imaging enable structural and microstructural evaluation of the brain in presymptomatic and symptomatic SCA2 gene carriers, in whom they show the typical pattern of olivopontocerebellar atrophy observed at neuropathological examination. Proton MR spectroscopy reveals, in the pons and cerebellum of SCA2 gene carriers, a more pronounced degree of abnormal neurochemical profile compared to other spinocerebellar ataxias with decreased NAA/Cr and Cho/Cr, increased mi/Cr ratios, and decreased NAA and increased mI concentrations. These neurochemical abnormalities are detectable also in presymtomatic gene carriers. Resting state functional MRI (rsfMRI) demonstrates decreased functional connectivity within the cerebellum and of the cerebellum with fronto-parietal cortices and basal ganglia in symptomatic SCA2 subjects. ^18^F-fluorodeoxyglucose Positron Emission Tomography (PET) shows a symmetric decrease of the glucose uptake in the cerebellar cortex, the dentate nucleus, the brainstem and the parahippocampal cortex. Single photon emission tomography and PET using several radiotracers have revealed almost symmetric nigrostriatal dopaminergic dysfunction irrespective of clinical signs of parkinsonism which are already present in presymtomatic gene carriers. Longitudinal small size studies have proven that morphometry and diffusion MR imaging can track neurodegeneration in SCA2, and hence serve as progression biomarkers. So far, such a capability has not been reported for proton MR spectroscopy, rsfMRI and NM techniques. A search for the best surrogate marker for future clinical trials represents the current challenge for the neuroimaging community.

## 1. Introduction

Spinocerebellar ataxia type 2 (SCA2) (OMIM 183090) is an autosomal dominantly inherited condition due to expansion of an unstable cytosine-adenineguanine (CAG) trinucleotide repeat within the coding sequence of the *SCA2* gene located at chromosome 12q24.1 and belongs to the polyglutamine (PolyQ) diseases group [1].

Clinical features of SCA2 include progressive neurological dysfunctions that present with either a cerebellar syndrome or, less frequently, Dopa-responsive Parkinson’s syndrome [2] (Table 1).

Gross neuropathological examination in brain of patients with genetically proven SCA2 reveals an olivo pontocerebellar atrophy (OPCA) with extension of the volume loss to the frontal lobes and spinal cord in advanced phases [3,4]. Histological examination demonstrates a distributed pattern of neuronal loss, which is particularly severe in thePurkinje neurons of the cerebellar cortex, gliosis and myelin loss [4,5].

Herein we review data concerning in vivo brain structure and microstructure evaluated with Magnetic Resonance (MR) imaging, neurochemical profiles assessed with proton MR spectroscopy, and functions explored with Blood Oxygenation Level Dependent (BOLD) MR contrast, so called functional MR Imaging (fMRI), and nuclear medicine (NM) techniques, in human carriers of SCA2 gene. To that end we conducted a systematic and comprehensive literature search to select relevant studies published up to September 2019 using the Medline database (Pubmed). The following keywords were selected: “Spinocerebellar ataxia 2”,“Spectroscopy spinocerebellar ataxia 2”, “Spectroscopy SCA2”, “MR spinocerebellar ataxia 2”,“MR SCA2”, “Functional MR spinocerebellar ataxia 2”,“Functional MR SCA2”, “SPECT spinocerebellar ataxia 2”,“SPECT SCA2”, “PET spinocerebellar ataxia 2”, and “PET SCA2.”

These neuroimaging data can serve as physiopathological, progression and surrogate biomarkers [6,7], whereas the wide availability of the molecular genetic test has considerably reducedthe diagnostic role of neuroimaging biomarkers in SCA2. In view of future therapeutic trials, particular attention will be paid to the studies investigating pre-symptomatic SCA2 carriers who, thanks to the autosomal dominant inheritance of the disease, can easily been recognized by the molecular genetic test in family members of affects patients. The pre-symptomatic subjects are in fact those for whom the major beneficial effects of new therapies for SCA2 are expected.

## 2. Structural and Microstructural MR Imaging

T1 weighted images show in SCA2 patients, the typical brainstem and cerebellar atrophy consistent with an OPCA pattern [7,8] (Figure 1). This is combined in proton density and T2 weighted images with diffuse hyperintensity of the cerebellar white matter, middle cerebellar peduncles and central pons with a characteristic sparing in the latter of the pyramidal tracts and medial lemnisci featuring a “cross” sign [9,10] (Figure 1). Differently from multiple system atrophy, the T2 signal in the striata is normal [9,11] (Figure 1).

Morphometry performed on T1-weighted images using volumetry of regions of interest or whole-brain analytical software, including voxel-based morphometry (VBM) or tensor-based morphometry, allows a quantitative assessment of the selective brainstem and cerebellar atrophy in SCA2 [12,13,14] (Figure 2) which is already present in pre-symptomatic gene carriers [15,16] (Figure 3).Atrophy of the right temporo-occipital cortex (parahippocampal, fusiform and lingual gyri) was observed in a meta-analysis of the VBM studies [14] (Figure 2).Striata and thalami have normal volumes [12,14].

Volumes of the brainstem and cerebellum showed inverse correlations with severity of clinical deficit [17], age of onset and CAG repeat length [16].

Analysis of fractal dimension has revealed that structural complexities of the cerebral and cerebellar cortex and white matter are decreased [18].

Diffusion MR imaging, using regions of interest, histogram analyses or tractography or tract-based spatial statistics, allows one to quantitatively measure the symmetric distributed microstructural damage of the T2 hyperintense or normal white matter in the brainstem, cerebellar peduncles, cerebellum and corticospinal tracts of SCA2 gene carriers [9,17,19,20,21,22,23] (Figure 4). Additional areas of increased diffusivity and decreased fractional anisotropy can be observed in the thalamus, corpus callosum and cerebral hemispheric white matter [17,20,21,23].

Generally, modifications of diffusion properties in the brain of SCA2 gene carriers correlated with ataxia severity [9,17,20,21], cognitive scores [22] and disease duration [17].

Morphometry [13,23] (Figure 5) and diffusion MR [23,24] in small size longitudinal studies documented progression of neurodegeneration in terms of accelerated volume loss and microstructural changes in SCA2 compared to age matched healthy control subjects. This proves their capability to serve as progression biomarkers. Notably, a greater sensitivity (effect size) of morphometry than clinical scores was reported in one study [23].

## 3. MR Spectroscopy

Proton MR spectroscopy allows the evaluation of selected volumes of interest regarding the relative or absolute amount of a variable number of metabolites depending on the strength of the magnetic field and on the sequence utilized [25,26]. These includeN-acetyl aspartate (NAA), which is an indicator of neuronal viability and integrity [27,28,29]; choline-containing compounds (Cho), which are indicators of membrane metabolism [27,28]; myo-Inositol (mI), which is a marker of glial activation [30], and therefore, of neuronal injury and degeneration [31]; and total creatine (Cr),which reflects the energy metabolism in the brain. The Cr concentration is assumed to remain stable in the healthy brain tissue with intact brain energy metabolism and it is often used as a reference for comparisons [26].

Despite the small sizes of the CNS structures, the CSF motion and the compact bone delimiting the posterior cranial fossa, all representing technical challenges for MR spectroscopy, proton MR spectra of adequate quality can be acquired in few minutes in most of the brainstem and cerebellum structures with the exception of the medulla [32].

Accordingly, several cross-sectional proton MR studies investigated the pons and dentate-peridentate or vermian cerebella in SCA2 gene carriers and healthy control subjects [19,33,34,35,36,37,38,39,40,41,42].

A recent meta-analysis of the proton MR spectroscopy studies in spinocerebellar ataxias [43] reported that SCA2 gene carriers showed the more pronounced abnormal neurochemical profiles in the pons and cerebellum compared to SCA1, SCA3, SCA6; and Friedreich’s ataxia with decreased NAA/Cr and Cho/Cr (Figure 1) and increased mI/Cr ratios. Decreased NAA and total NAA concentrations [19,34,41,42], increased total Cr concentration [34,36,41,42] and increased mI concentration [34,41,42] have also been reported in the same structures, as has the presence of lactate in the cerebellum [33,35]. Two studies at high field [41,42] reported a decrease of glutamate concentration, a marker of neuronal function, in the pons and cerebellum.

Correlations between the neurochemical abnormality and severity of a clinical deficit [35,38,39,41,42], quality of life [42], CAG repeats [34,39], disease duration [34,42] and age at onset of symptoms [34] were reported.

One study [34] investigated additional brain structures in SCA2 gene carriers and healthy controls and reported a decrease of the NAA/Cr and increase of the mI/Cr ratio in the frontal cortex and basal ganglia.

Importantly, in a recent study, premanifest SCA2 mutation carriers with estimated disease onsets within 10 years had decreased total NAA/mI ratios in the dentate and peridentate region, cerebellar vermis and pons that were in the range of early manifest SCA subjects [42].

In the only available longitudinal proton MR spectroscopy study [40] in five SCA2 patients, no statistically significant differences were observed for NAA/Cr and NAA/Cho ratios between the initial and follow-up examinations performed a mean of 38 months apart.

## 4. Functional MRI

MRI exploiting BOLD contrast is a major tool to non-invasively investigate the brain function in patients with neurological diseases. As in other neurodegenerative diseases [44], clinical impairment hinders assessment of symptomatic SCA2 gene carriers with fMRI during execution of tasks [45], and justifies that resting state fMRI has been the preferred method to investigate brain physiopathology in SCA2 using a variety of analytical approaches [46,47,48,49].

In symptomatic SCA2, subjects presenting with ataxia were reported to have decreased connectivity within the cerebellum and between the cerebellum and frontal-parietal cortices [47,48], with a selective decrease of the connectivity of the anterior regions of the cerebellum with the somatosensory cortex and of the posterior regions of the cerebellum with cortical cerebral regions related to cognition and emotion [49] (Figure 6). An increased connectivity of the cerebellum with parietal, frontal and temporal areas was also reported in SCA2 patients in one study [48].

In SCA2 patients presenting with parkinsonism, Wu et al. [46] observed decreased connectivity within the basal ganglia-thalamus-cortical motor loop, and cortico-cortical motor, cortico-cerebellar and cortico-brainstem circuits compared to healthy controls, which were normalized after levodopa treatment. Notably, in pre-symptomatic SCA2 gene carriers of the same families, the connectivity within the basal ganglia-thalamus-cortical motor loop was decreased, but the connectivity of the remainder circuits was increased compared to controls.

One study reported significant correlations between clinical scores and the abnormal functional connectivity strength [48].

No longitudinal study of fMRI in SCA2 gene carriers has been published so far.

## 5. Nuclear Medicine

Positron Emission Tomography (PET) and Single Photon Emission Computed Tomography (SPECT) using different radiotracers enable a multi-domain functional evaluation of the brain and cerebellum that, in comparison to fMRI, has the main drawback of requiring radiation exposures [50,51].

### 5.1. Glucose Metabolism

All the small size studies that used PET and^18^F-fluorodeoxyglucose demonstrated severely decreased metabolism in the cerebellar cortex, with preferential involvement of the anterior lobe, the dentate nucleus and brainstem in symptomatic SCA2 gene carriers [52,53,54] (Figure 7); that was comparatively more pronounced than in SCA3 and SCA6 patients in one study (53). Decreased glucose uptake was observed also in the parahippocampal, frontal and parietal cortices [52,53]. Pre-symptomatic carriers of SCA2 mutation already show hypometabolism in the pons and cerebellum [55].

The small sample sizes hindered clinical correlation of this metabolic abnormality in the available studies.

### 5.2. Nigrostriatal System

Almost symmetric nigrostriatal dysfunction secondary to substantia nigra degeneration with decreased striatal uptake of specific tracers is an established feature of SCA2 in symptomatic subjects presenting with cerebellar syndrome or parkinsonism. It has been documented using SPECT and dopamine active transporter (DAT) tracers [56,57,58,59] and using PET and both 6-[^18^F]fluoro-L-dopa and positron emitting DAT tracers [52,60,61,62] (Figure 8). It has also been reported in pre-symptomatic subjects [59]. A voxel-wise analysis revealed that decreased DAT tracer uptake can also be appreciated in the midbrain and pons in SCA2 patients [63]. In line with the responsiveness to levodopa treatment in SCA2 patients with clinical evidence of parkinsonism, striatal D2 receptors are not affected [61,62]. The intriguing coexistence of severe nigrostriatal dysfunction on NM and lack of overt clinical signs of parkinsonism in the majority of SCA2 patients has been explained in a combined NM and neuropathological study by the demonstration of a severe neurodegeneration in the subthalamic nuclei, which, as is observed in clinical cases of amelioration of extrapyramidal sings by stereotactic lesion or deep brain stimulation of these nuclei in idiopathic Parkinson’s disease, would counteract the effects of severe degeneration of the substantia nigra [62].

Longitudinal NM studies are not available in SCA2.

## 6. Limitations and Perspectives

Rarity of SCA2 justifies the small number of SCA2 gene carriers recruited in single center neuroimaging studies, and, along with heterogeneity of methods, may explain some degree of variability of the results obtained so far. Two solutions have been proposed to overcome this limitation.

Meta-analyses of different single studies, such as those recently published by Han et al. [14] for voxel-based morphometry studies and by Krahe et al. [43] for proton MR spectroscopy studies.Data pooling with centralized analyses or re-analyses, as in the European Integrated Project on Spinocerebellar Ataxias (EUROSCA) (www.eurosca.org), the study of individuals at risk for SCA1, SCA2, SCA3 and SCA6 (RISCA) [15] and the ENIGMA project (http://enigma.ini.usc.edu/ongoing/enigma-ataxia).

Neuroimaging techniques have been advocated as surrogate markers [6] for the next coming clinical trials of spinocerebellar ataxias driven by advancements in genetic and molecular understanding as well as in pharmacological [64] and neurophysiological treatments [65,66]. In particular, due to the relative insensitivity of clinical scales [67] and the possibility of a “ceiling effect” hindering reversal of severe clinical deficits in patients with advanced disease, inclusion of neuroimaging data as surrogate markers would theoretically allow for conducting proof-of-concept and efficacy studies with smaller numbers of patients [6].

Although until now the use of neuroimaging techniques as surrogate markers in patients with chronic ataxias has been limited to few exploratory, observational, single center studies that recruited few patients [7], some specific considerations on SCA2 are in order.

Converging data from different neuroimaging techniques [12,43,53] indicate that the severity of neurodegeneration in SCA2 featuring an OPCA pattern exceeds levels of severity in other SCAs. This implies that SCA2 gene carriers may deserve priority as targets for proof-of concept therapeutic studies in humans with spinocerebellar ataxias.

Structural (volumes) and microstructural data have already demonstrated their capability to track progression of neurodegeneration in SCA2 [13,23,24] and this candidate’s T1-weighted MR images and DTI as potential surrogate biomarkers. However, the scarce elastic properties of volumetry makes it not very attractive for such a purpose. In fact, only a modification of the slope of volume loss over time can conceivably be expected in responders to therapy compared to untreated control patients. Hence greater expectations as surrogate neuroimaging biomarkers are being turned to DTI, and especially, proton MR spectroscopy, resting state fMRI and NM [7,41,42]. However, the sensitivity and robustness of each of these neuroimaging methods in this context have not yet proven. The search for the best surrogate marker for futureclinical trials in SCAs represents the current challenge for the neuroimaging community.

## 7. Conclusions

Neuroimaging techniques explore brain structure and microstructure with morphometry and diffusion MRI, the neurochemical profile with proton MR spectroscopy, the functional connectivity with resting state fMRI; and metabolism and nigrostriatal function with PET and SPECT. These quantitative features are valuable physiopathological biomarkers in SCA2 symptomatic and asymptomatic gene carriers. Morphometry and diffusion MRI allow tracking neurodegeneration. Although proton MR spectroscopy, resting state fMRI and PET and SPECT are expected to be more sensitive biomarkers of disease progression, this has not been proven so far. Overcoming the small size typical of single center neuroimaging studies for a rare disease, SCA2 studies are undergoing data pooling in specific initiatives. A selection of the best surrogate biomarker represents the current challenge of the neuroimaging community in view of coming therapies.

## Figures and Tables

**Figure 1 ijms-21-01020-f001:**
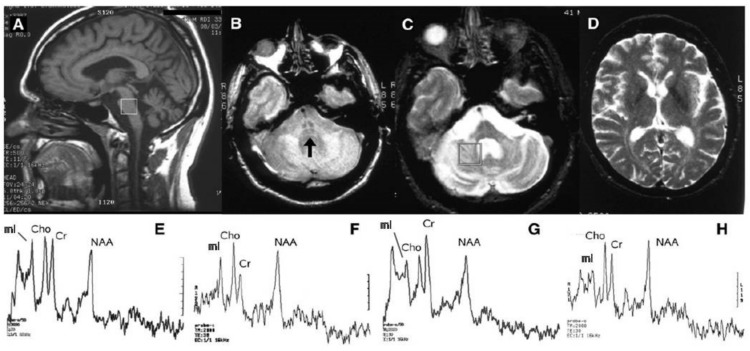
MR imaging and proton MR spectroscopy in a symptomatic 41-year-old man with SCA2. Sagittal T1 weighted (**A**), axial proton density (**B**) and T2 weighted (**C**) images show a pattern consistent with olivopontocerebellar atrophy with evidence of the “cross sign” (arrow) in (**B**). The T2 signal in the basal ganglia (**D**) is normal. Proton MR spectroscopy (STEAM TR 2000 ms TE 30 ms) of 8 mL single voxels placed in the basis pontis (white square in A) of the SCA2 patient (**E**) and of a healthy control (**F**), and in the deep right cerebellar hemisphere (white square in C) of the SCA2 patient (**G**) and of the healthy control (**H**), show lower NAA/Cr, Cho/Cr and NAA/mI ratios in the SCA2 patient in both locations. Modified and reproduced with permission fromMascalchi M. and Vella A.: Magnetic resonance and nuclear medicine imaging in ataxias.Handb Clin Neurol 103: 85–110, 2012.

**Figure 2 ijms-21-01020-f002:**
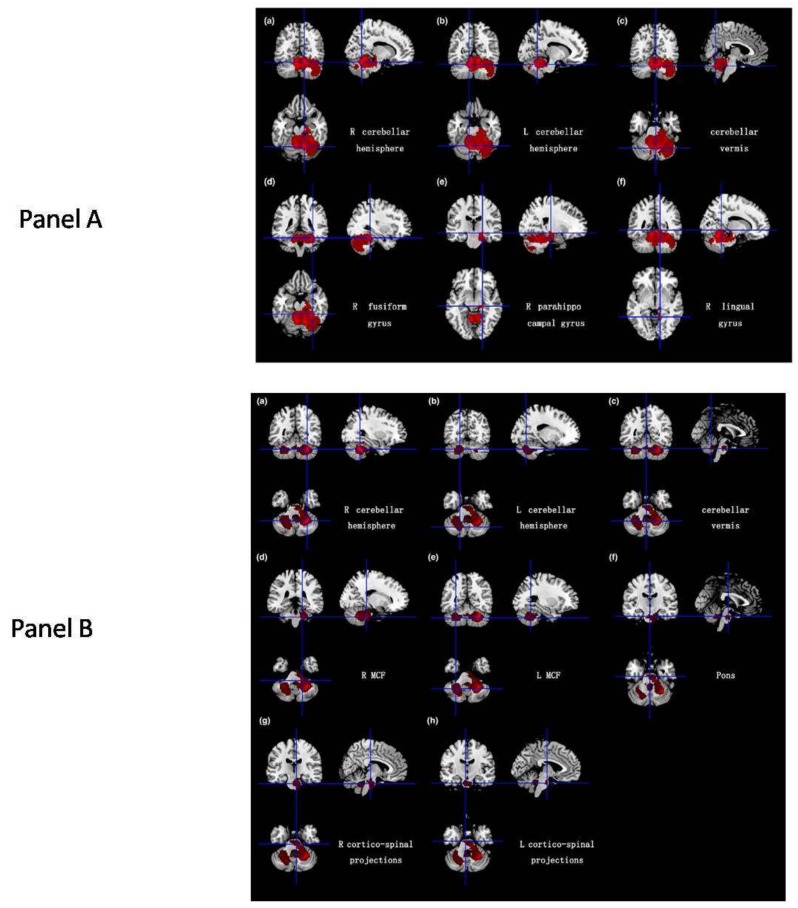
(**A**,**B**).Results of a meta-analysis of 5 VBM studies in 65 SCA2 gene carriers and 124 healthy controls. Panel A shows atrophy (in red) of the gray matter in bilateral cerebellar hemispheres, cerebellar vermis, the right fusiform gyrus, right parahippocampal gyrus and the right lingual gyrus. Panel B shows the atrophy (in red) of the white matter in bilateral cerebellar hemispheres, cerebellar vermis, middle cerebellar peduncles, pons and bilateral cortico-spinal projections. Reproduced from Han Q, Yang J, Xiong H and Shang H: Voxel-based meta-analysis of gray and white matter volume abnormalities in spinocerebellar ataxia type 2. Brain Behavior 8: e01099, 2018.

**Figure 3 ijms-21-01020-f003:**
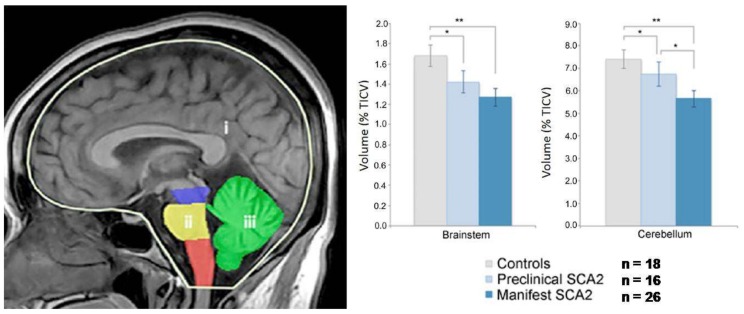
Volumetry in preclinical and manifest SCA2 gene carriers. The volumes of the brainstem (ii) comprising the mesencephalon (blue), pons (yellow) and medulla oblongata (orange), and the cerebellum (iii) (green), both normalized to the total intracranial volume (TICV) (i) (left panel), are significantly (** = *p* < 0.0001 and * = *p* < 0.05) lower in manifest and preclinical SCA2 gene carriers than in controls (right panel). Modified fromReetz K., Rodríguez-Labrada R., Dogan I, Mirzazade S., Romanzetti S., Schulz J.B., Cruz-Rivas E.M., Alvarez-Cuesta J.A., Aguilera Rodríguez R., Gonzalez Zaldivar Y., et al.: Brain atrophy measures in preclinical and manifest spinocerebellar ataxia type 2. Ann Clin Transl Neurol 5:128–137, 2018.

**Figure 4 ijms-21-01020-f004:**
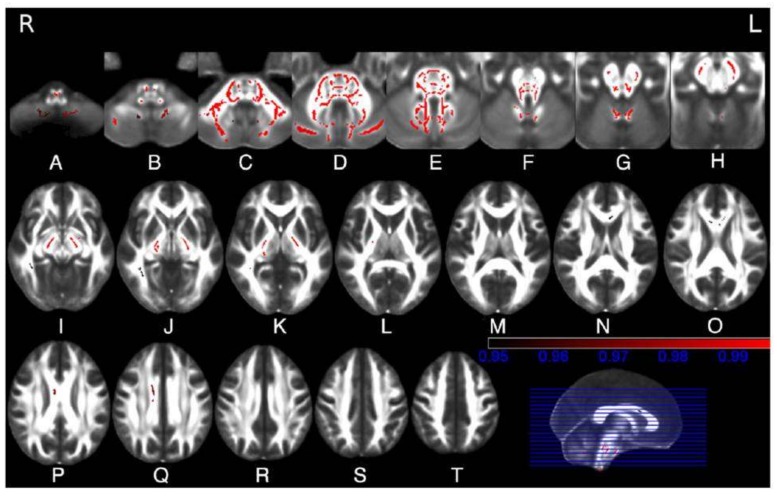
(**A**–**T**). Tract-basedspatial statistical analysis of diffusion tensor imaging data in 10 SCA2 patients vs. 10 healthy controls. Maps show in red the clusters of significantly reduced fractional anisotropy in white matter tracts in the SCA2 patients. These include the inferior (**B**,**C**), middle (**C**,**D**) and superior (**E**,**F**) cerebellar peduncle, the cerebellar white matter (**C**,**D**,**E**,**F**), the medial and lateral lemnisci and the spinothalamic tracts (**D**,**E**,**F**), the transverse pontine fibres (**D**,**E**), the corticospinal tracts at the level of the internal capsule (**J**,**K**,**L**), cerebral peduncles (**G**,**H**,**I**), basis pontis (**C**,**D**) and bulbar pyramis (**A**,**B**), the corpus callosum (**N**,**O**,**P**,**Q**), the right inferior longitudinal fasciculus (**I**,**J**) and the inferior fronto-occipital fasciculus (**I**,**J**). Reproduced with permission from Della Nave R., Ginestroni A., Tessa C., Salvatore E., De Grandis D, Plasmati R, Salvi F, De Michele G, Dotti MT, Piacentini S, et al.: Brain white matter damage in SCA1 and SCA2. An in vivo study using voxel-based morphometry, histogram analysis of mean diffusivity and tract-based spatial statistics. NeuroImage 43: 10–19, 2008.

**Figure 5 ijms-21-01020-f005:**
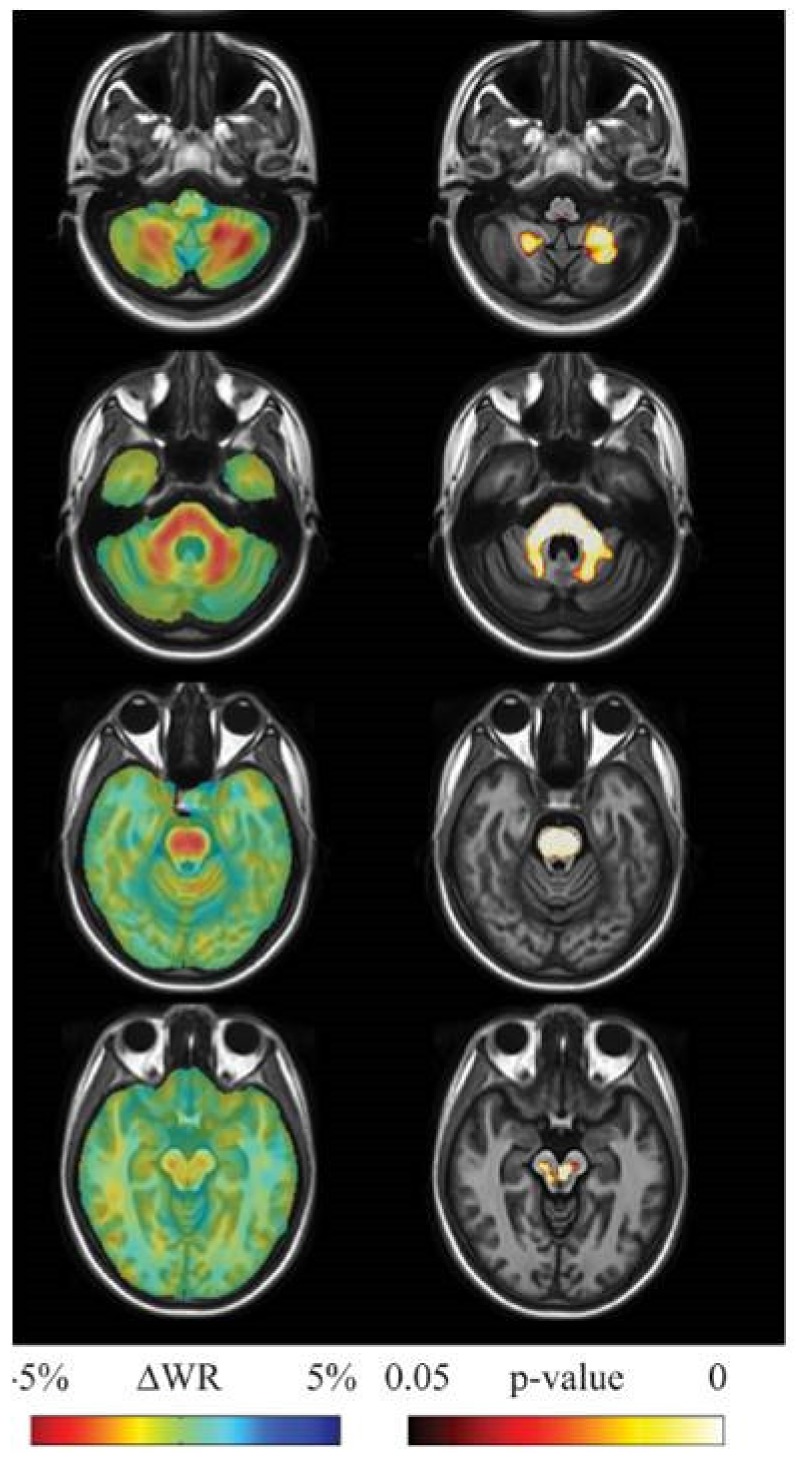
Results of longitudinal between group (10 SCA2 vs. 16 healthy controls, each examined with a mean interval of 3.3 years between initial and follow-up MRI) tensor-based morphometry analysis. Left panel: sample of axial views of the difference in average longitudinal warp rate (WR) maps between SCA2 patients and healthy controls, where red indicates local thinning and blue indicates local enlargement. Right panel: voxel-wise threshold-free cluster enhancement corrected *p*-value maps at the same levels, testing the null hypothesis of zero differences in WR between SCA2 patients and healthy controls. Highlighted clusters indicate significantly (*p*<0.05) accelerated volume loss in SCA2 patients when compared to healthy controls. SCA2 patients exhibit significantly accelerated volume loss in the midbrain (substantia nigra and medial lemniscus, bilaterally, right lateral lemniscus and central region corresponding to decussation of the superior cerebellar peduncles), the entire basis pontis, the middle cerebellar peduncles and posterior medulla corresponding to the gracilis and cuneatus tracts and nuclei. The cerebellum shows accelerated loss of the white matter in the hemispheric and peridentate regions and of the gray matter in the cerebellar cortex of the inferior portions of the cerebellar hemisphers. Reproduced from Mascalchi M, Diciotti S, Giannelli M, Ginestroni A, Soricelli A, Nicolai E, Aiello M, Tessa C, Galli L, Dotti MT, et al.: Progression of brain atrophy in SCA2. A longitudinal TBM study.PLoS One25; 9(2):e89410, 2014.

**Figure 6 ijms-21-01020-f006:**
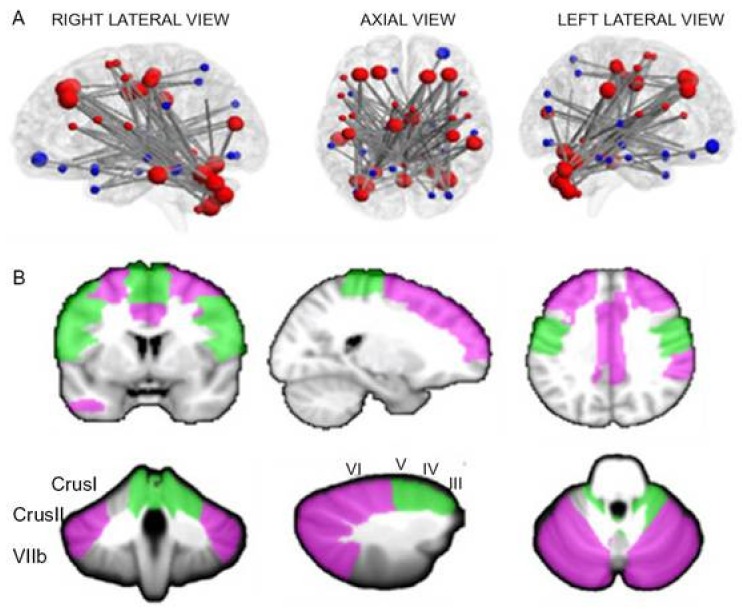
(**A**,**B**). Panel A shows the network of significantly decreased functional connectivity in 9 SCA2 patients compared to 33 healthy controls as assessed by network-based statistics analysis. The regions of the cerebello-cortical (red) and cortico-cortical (blue) modules are shown in different colors. Bigger nodes correspond to cerebellar and cortical regions relevant to cognition and emotion; smaller nodes correspond to cerebellar and cortical regions relevant to motor control. Panel B shows the anatomical representations of cognitive (violet) and motor (green) nodes in the cerebellum and cerebral cortex showing underconnectivity between each other. Reproduced from Olivito G, Cercignani M, Lupo M, Iacobacci C, Clausi S, Romano S, Masciullo M, Molinari M, Bozzali M, Leggio M:. Neural substrates of motor and cognitive dysfunctions in SCA2 patients: A network-based statistics analysis. NeuroImage Clin 14:719–725, 2017.

**Figure 7 ijms-21-01020-f007:**
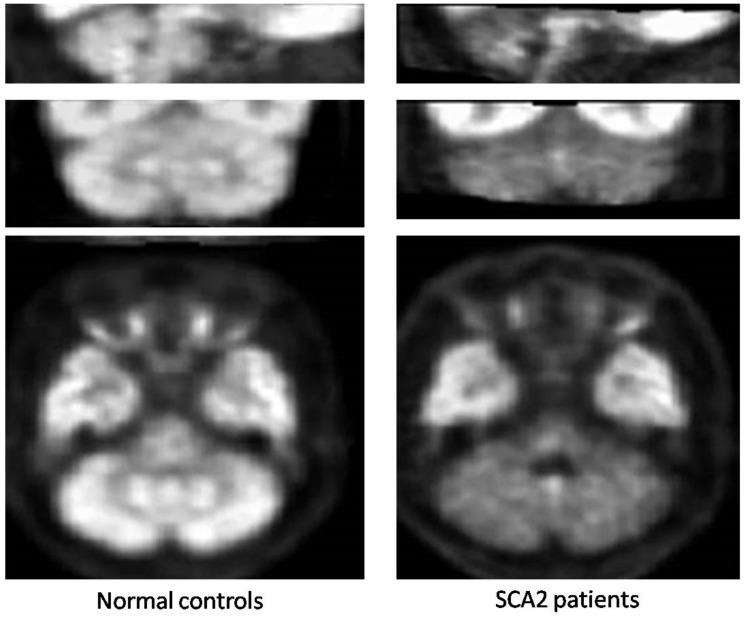
18F-fluorodeoxyglucose positron emission tomography in SCA2.Representative sagittal (top), coronal (mid) and axial (bottom) images of spatially normalized ratios of cerebellar to cerebral 18F-fluorodeoxyglucose uptake using study-specific templates in 89 normal controls (left panels) and 9 patients with SCA2 (right panels) show diffusely decreased hypometabolism in the cerebellar cortex and dentate nuclei (mid and bottom images). Note in the SCA2 patients, the decreased tracer uptake in the basis pontis and medulla oblongata as well (top and bottom images). Modified from Oh M, Kim JS, Oh JS, Lee CS, Chung SJ: Different subregional metabolism patterns in patients with cerebellar ataxia by 18F-fluorodeoxyglucose positron emission tomography. PLoS One 12: e0173275, 2017.

**Figure 8 ijms-21-01020-f008:**
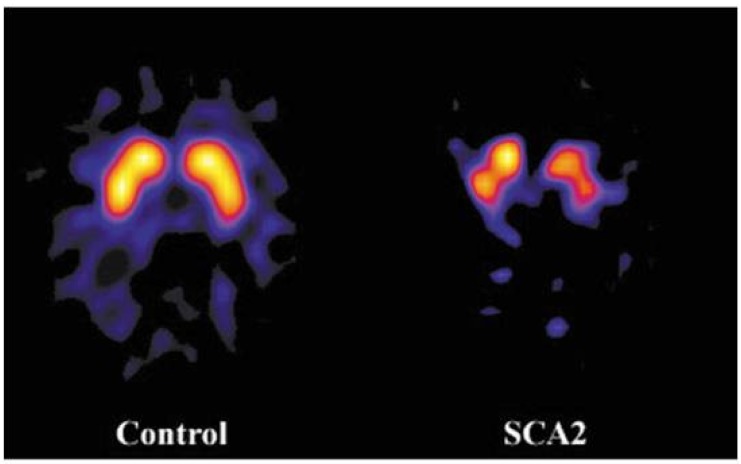
Axial [123I]FP-CIT single photon emission computed tomography scans at the level of the basal ganglia show marked and uniform reduction of the dopamine active transporter (DAT)tracer uptake in a symptomatic SCA2 gene carrier without parkinsonism (right) compared to a control subject (left).Modified and reproduced with permission from Varrone A, Salvatore E, De Michele G, Barone P, Sansone V, Pellecchia MT, Castaldo I, Coppola G, Brunetti A, Salvatore M, et al.: Reduced striatal [123 I]FP-CIT binding in SCA2 patients without parkinsonism. Ann Neurol 55: 426–430, 2004.

**Table 1 ijms-21-01020-t001:** Signs and Symptoms of Spinocerebellar ataxia type 2.

Common (Adult Phenotype)
Progressive cerebellar ataxia
Dysarthria
Dysphagia
Oculomotor dysfunction
Pyramidal signs
Signs of lower motor neuron degeneration
Extra-pyramidal features *
Sensory-motor peripheral neuropathy
Painful muscle cramps
Autonomic dysfunction
Olfactory deficit
Sleep disturbances
Cognitive decline
Psychiatric symptoms
**Uncommon (Infantile Phenotype)**
Developmental delay
Facial dysmorphism
Retinitis pigmentosa
Myoclonus-epilepsy

* Early-onset L-dopa-responsive parkinsonism can represent clinical presentation in some families.

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
