# Peer review of "Neuroimaging Biomarkers in SCA2 Gene Carriers"

_ijms, 2020, doi:10.3390/ijms21031020_

Round 1
Reviewer 1 Report
In the present paper, the Authors reviewed data concerning in vivo brain structure and microstructure evaluated with Magnetic Resonance (MR) imaging, neurochemical profile assessed with Proton MR spectroscopy and function explored with Blood Oxygenation Level Dependent (BOLD) MR contrast, so called functional MR Imaging (fMRI), and nuclear Medicine (NM) techniques in human carriers of SCA2 gene. These neuroimaging data may serve as physiopathological, progression and surrogate biomarkers, whereas the wide availability of the molecular genetic test has considerably reduced the diagnostic role of neuroimaging biomarkers in SCA2.
Overall, I found the present Review very interesting, well written and structured, timely and scientificallu sound: enjoyed reading it!
I have only some minor comments aimed to improve the high quality of the paper and these are explained below:
1) I suggest to add a smal table with signs and symptoms of Spinocerebellar ataxia type 2 (SCA2) that would be useful for the reader.
2) I suggest Authors to add a brief paragraph on how literature search was conducted (i.e. a brief paragraph on Methods of the review).
Reviewer 2 Report
In this review article the authors presents a nice overview of the neuroimaging biomarkers for SCA2 gene carriers. In particular, findings obtained by using magnetic resonance imaging and nuclear medicine methods in these patients are described in detail.
The manuscript is well-structured and organized.
I would suggest minor revisions:
- please verify that along the text and the abstract the terms PET and SPECT are accompanied by the radiotracer used.
- please verify that the articles reported in the reference list are written according to the journal' guideline.
Author Response
Please see the attachment (also for Reviewer 1)
